# *Beet Curly Top Iran Virus* Rep and V2 Suppress Post-Transcriptional Gene Silencing via Distinct Modes of Action

**DOI:** 10.3390/v15101996

**Published:** 2023-09-26

**Authors:** Saeideh Ebrahimi, Omid Eini, Alexandra Baßler, Arvid Hanke, Zeynep Yildirim, Michael Wassenegger, Gabi Krczal, Veli Vural Uslu

**Affiliations:** 1RLP AgroScience GmbH, Breitenweg 71, 67435 Neustadt an der Weinstraße, Germany; 2Department of Plant Protection, University of Zanjan, Zanjan 313, Iran; 3Department of Phytopathology, Institute for Sugar Beet Research, 37079 Göttingen, Germany; 4MAPS, COS, Heidelberg University, 69120 Heidelberg, Germany

**Keywords:** BCTIV, VSR, Rep, V2, RNAi, RDR6, PTGS, *Nicotiana benthamiana*, geminivirus

## Abstract

*Beet curly top Iran virus* (BCTIV) is a yield-limiting geminivirus belonging to the *becurtovirus* genus. The genome organization of BCTIV is unique such that the complementary strand of BCTIV resembles *Mastrevirus*, whereas the virion strand organization is similar to the *Curtovirus* genus. Geminiviruses are known to avoid the plant defense system by suppressing the RNA interference mechanisms both at the transcriptional gene silencing (TGS) and post-transcriptional gene silencing (PTGS) levels. Multiple geminivirus genes have been identified as viral suppressors of RNA silencing (VSR) but VSR activity remains mostly elusive in becurtoviruses. We found that BCTIV-V2 and -Rep could suppress specific Sense-PTGS mechanisms with distinct efficiencies depending on the nature of the silencing inducer and the target gene. Local silencing induced by GFP inverted repeat (IR) could not be suppressed by V2 but was partially reduced by Rep. Accordingly, we documented that Rep but not V2 could suppress systemic silencing induced by GFP-IR. In addition, we showed that the VSR activity of Rep was partly regulated by RNA-dependent RNA Polymerase 6 (RDR6), whereas the VSR activity of V2 was independent of RDR6. Domain mapping for Rep showed that an intact Rep protein was required for the suppression of PTGS. In summary, we showed that BCTIV-Rep and -V2 function as silencing suppressors with distinct modes of action.

## 1. Introduction

*Beet curly top Iran virus* (BCTIV) has a monopartite single-stranded DNA (ssDNA) genome and belongs to the *Becurtovirus* genus of the *Geminiviridae* family [1,2]. BCTIV is a yield-limiting virus, causing a serious leaf curl disease in economically important crops such as sugar beet, tomato, pepper, and beans [1,2]. The leaf curl symptoms induced by BCTIV are very similar to those caused by other members of the *Geminiviridae*, such as *Beet curly top virus* (BCTV) of the *Curtovirus* genus [3]. However, BCTIV transmission relies on a distinct vector, *Circulifer haematoceps* [4], and the genome organization of BCTIV differs from the other members of the *Geminividae* [1]. Yet the functions of the BCTIV-encoded proteins have not been addressed rigorously.

Geminiviruses encode four to eight proteins in the double-stranded DNA (dsDNA) replicative intermediate of their mono- or bi-partite genomes. The transcription from the geminivirus replicative intermediate takes place in a bidirectional fashion: complementary (C) strand and virion (V) strand [5]. While most geminiviruses contain a TAATATT/AC sequence at the replication origin (*ori*), *Becurtoviruses* have TAAGATT/CC. Geminivirus replication is initiated at *ori* by the viral replication initiator protein, known as the complementary strand protein 1 (C1) [5]. Other proteins in the complementary strand are transcriptional activator protein C2, replication enhancer protein C3, and pathogenicity determinant protein C4 [6]. In addition, in the *Mastrevirus*, *Capulavirus*, and *Grablovirus* genera, due to a splicing event within C1, a new protein called Rep emerges containing the N terminus of C1, a frame-shifted C terminus of C1, a 14 nt intergenic region, and C2 proteins. Rep protein of geminiviruses is a multifunctional protein, which is Indispensable for viral genome replication [7]. The virion strand encodes the coat protein (V1), movement protein (V2), and a poorly characterized gene (V3) [8]. The C and V genes are organized in unique combinations for each genus of the *Geminiviridae* family. For example, BCTIV contains C1, C2, and Rep on the complementary-sense strand, similar to *Mastreviruses*, whereas V1, V2, and V3 are on the virion-sense strand, similar to curtoviruses [9,10] (Appendix A).

Virus pathology is influenced by the efficiency of virus accumulation, which predominantly depends on viral suppression of RNA interference (RNAi) in plants [11]. RNAi is a conserved defense mechanism, acting through transcriptional (TGS) or post-transcriptional (PTGS) repression of gene expression in a sequence-specific manner [12]. Bi-directional transcription from the geminivirus genome creates double-stranded RNA (ds-RNA) [13]. In addition, aberrant geminivirus transcripts can be used as templates for dsRNA production by plant RNA-dependent RNA polymerases (RDRs) [14]. dsRNAs are cleaved into small interfering RNAs (siRNAs) of distinct lengths by Type III endonucleases, called DICER-LIKE (DCL) proteins [12]. siRNAs bind to ARGONAUTE (AGO) proteins and form the RNA-induced silencing complex (RISC) for the cleavage of the target sequence. Besides cleavage, RISC can also control PTGS by recruiting RDRs for amplification of the RNAi or inhibiting target gene translation facultatively [15,16,17]. In addition, auxiliary RNAi proteins like HUA ENHANCER 1 (HEN1) RNA methyltransferase, which protects siRNAs from degradation, contribute to viral defense in plants [18,19]

Many viral PTGS suppressors have been identified in plant viruses [20]. For example, helper component-proteinase (HC-Pro) encoded by *Potyviridae* inhibits RDR6 [21] and HEN1 [22,23] among other components of the RNAi pathway, whereas, P19 protein (P19) of *Tomato bushy stunt virus* (TBSV) sequesters ds-siRNAs, thus inhibiting RNAi based protection in plants [24,25]. In addition, the P6 protein of *Strawberry vein banding virus* [26], the P20, P23, and capsid proteins of *Citrus tristeza virus* [27], the P6 protein of *Cauliflower mosaic virus* [28], the P38 protein of *Turnip crinkle virus* [29], and the P1 of *Sweet potato mild mottle virus* [30] are among those which were shown to function as viral suppressors of RNA silencing (VSR). In *Geminiviridae*, several reports implied that C2, C4, Rep, V2, and V3 proteins function as VSRs, involved in the suppression of transcriptional gene silencing (TGS) or post-transcriptional gene silencing (PTGS) in plants [5,11,31,32]. Yet, the silencing suppressor activity of the BCTIV gene products remains elusive.

In this study, we induced PTGS on a GFP transgene using specific silencing inducers, including full-length GFP, partial sense-strand GFP, and inverted-repeat GFP. By administering combinations of the given silencing inducers with each of the BCTIV open reading frame products on *Nicotiana benthamiana*, we investigated the efficiency of PTGS in local and systemic contexts to find out the BCTIV genes, which are responsible for PTGS suppression and deduce their modes of action.

## 2. Materials and Methods

### 2.1. Plasmid Constructs for Transient Expression

Sense-PTGS silencing inducers were cloned into pGreen (pG104) vectors containing full-length (GFP-FL) or only 139 nt long 5′-GFP sequence (5′-GFP) [33]. Strong PTGS was induced by the inverted repeat of the identical 139 nt sequence (GFP-IR) with a 90 nt long spacer as described in [33]. The positive VSR control P19 of TBSV was expressed in a pPCV702SM binary vector [34]. All silencing inducers were expressed under the control of the 35S promoter.

The genes on the virion and complementary strand of the BCTIV replication intermediate (C1, C2, Rep, V1, V2, V3) based on Kaftarak isolate (GenBank: KP410285.1) were expressed via pDIVA binary vectors [35] for transient expression on GFP-expressing *N. benthamiana* 16C plants (Appendix A). The coding sequences of the C1, C2, Rep, and V2 genes were amplified with primers containing *BamHI* and *XhoI* restriction sites, using Q5 DNA polymerase (NEB). The PCR products were first cloned into a pJET1.2/blunt cloning vector (Thermo Fischer Scientific, Karlsruhe, Germany) and they were sub-cloned using the *Xho*I and *BamHI* sites of a pG104 vector to obtain pG-C1, pG-C2, pG-Rep, and pG-V2. The inserts were confirmed by Sanger sequencing. We confirmed that the vector backbone did not influence the silencing or VSR phenotypes.

Twelve truncations from the Rep gene (Appendix A) were PCR-amplified using corresponding primers (Appendix A) and cloned into pJET1.2. Then, the plasmids with Rep truncations were digested with the *BamHI* and *XhoI* enzymes and the fragments were cloned into the pG104 vector backbone to obtain pG-Rep1, pG-Rep2, pG-Rep3, pG-Rep4, pG-Rep5, pG-Rep6, pG-Rep7, pG-Rep8, pG-Rep9, pG-Rep10, pG-Rep11, and pG-Rep12 (Appendix A). *Escherichia coli* (strain INVαF, ThermoFisher) was used for cloning plasmid constructs, and *Agrobacterium tumefaciens* strain ATHV was used for the transient transformation of the plants. A pSoup helper plasmid was used together with pG104 for transforming *Agrobacterium* by electroporation. In addition to the plasmid constructs obtained from INVαF, plasmids in the *Agrobacterium* clones were also sequenced to ensure the insert integrity (Appendix B).

### 2.2. Agroinfiltration

The BCTIV genes were co-infiltrated with *Agrobacterium* ATHV containing silencing inducer constructs into wildtype *Nicotiana benthamiana* (WT) and also GFP-expressing *N. benthamiana* (16C) at the four/six-leaf-stage. *Agrobacterium* containing the constructs were grown in LB medium (containing Kanamycin and Rifampicin) at 28.0 °C for 16 h with continuous shaking at 135 rpm, then re-suspended in MES buffer (10 mM MgCl_2_, 10 mM MES (pH 5.6), and 100 μM acetosyringone) to reach OD_600_ = 1.0 for agroinfiltration. For the negative control, *N. benthamiana* plants were separately infiltrated with FL-GFP, 5′-GFP, and GFP-IR. In order to equalize the strength of silencing induction in single silencing inducer infiltrations, *Agrobacterium* containing empty pG104 (OD_600_ = 1) was added for mock co-infiltrations. Unless otherwise stated, three plants were used for each treatment in each experiment, and each experiment was repeated at least three times at different times of the year to avoid any effect of temperature fluctuations in the greenhouse.

### 2.3. GFP Imaging and Quantification

The inoculated plants were maintained in a growth chamber at 25 °C with 16 h light/8 h dark periods. After 3 days post inoculation (dpi), the control plants were monitored regularly under UV light (Blak-Ray B-100 AP Lamp, Analytik Jena, Jena, Germany) to visualize the silencing phenotype. At 6 dpi all the plants were recorded using a camera (Canon EOS700D, 18–55 mm Lens, Canon, Mannheim, Germany) under UV light and leaves were scanned using a laser scanner (Molecular Imager^®^ PharosFX™ Systems, BioRad, Hercules, CA, USA) at 50 µm/pixel, in GFP (excitation at 488 nm, emission at 530 nm) and chlorophyll (excitation at 488 nm, emission at 695 nm) channels.

The GFP and the chlorophyll images were merged and the GFP expression was quantified in the limited area of infiltrated patches. GFP expression was measured by FIJI image analysis in treated, non-necrotic areas, and the acquired values were normalized using non-treated areas. The final values were plotted in bar charts and mean value graphs together with the actual data points.

### 2.4. Protein Extraction and Western Blot

Total protein was extracted from infiltrated leaf patches of WT plants 6 dpi using protein extraction buffer (1M Tris-HCl, pH 8.8, 10% Glycerol, 10% SDS, and H_2_O) together with cOmplete^™^ Protease inhibitor solution (Roche, Basel, Switzerland). SDS-PAGE was performed for Coomassie staining and Western blotting, and polyclonal goat anti-GFP antibody (AB0066-200, 1:1000, Sicgen, 3060-197, Cantanhede, Portugal) as primary, donkey anti-goat IgG-HRP antibody (sc-2020, Santa Cruz Biotechnology (Dallas, TX, USA), (1:5000) as secondary antibody were used for Western blot. Pierce™ ECL Western Blotting Substrate (ThermoFisher) was used to detect the signal.

### 2.5. RNA Extraction, RT-PCR, and Northern Blot

Total RNA was extracted from infiltrated patches of 16C leaves at 6 dpi using TRIzol^®^ Reagent (Qiagen, Hilden, Germany). RNA concentration and RNA purity were measured using Nanodrop (ND-1000) and RNA quality was assessed by agarose gel electrophoresis. RNAs were treated with Lucigen Baseline-ZERO DNase and purified by acid-Phenol-Chloroform pH:4.5 (with Isoamyl alcohol, 25:24:1) (Invitrogen™, Thermofisher, Karlsruhe, Germany). RT-PCR was performed using the SuperScript™ III One-Step RT-PCR System with Platinum™ Taq DNA Polymerase (Invitrogen™, ThermoFisher, Karlsruhe, Germany) and gene-specific primers (Appendix A) to amplify the BCTIV genes to confirm transient expression of the constructs upon agroinfiltration. For the Northern blot of GFP mRNA and GFP siRNA, 10 μg and 5 μg of total RNA were used, respectively, as previously described [33]. The siRNA Northern blots were stripped and rehybridized with a U6 snoRNA probe. For the GFP probes (alpha-P32) CTP was used for labelling, and for the U6 snoRNA probe (gamma-p32) ATP was used from Hartman Analytic, Germany. The detection of the Northern blot signals was performed by the Molecular Imager^®^ PharosFX™ Systems (BioRad). qRT-PCR was performed by Luna Universal One-Step RT-qPCR Kit (E3005, New England Biosciences (NEB),Frankfurt, Germany) using cDNA, synthesized by ProtoScript^®^ First Strand cDNA Synthesis Kit (E6300, NEB, Frankfurt, Germany) using the primers (Appendix A).

### 2.6. Rdr6 N. benthamiana Mutant Lines

Th homozygous *rdr6* line, which contains a single nucleotide insertion that leads to a frameshift in RNA Dependent RNA Polymerase 6 (RDR6), was kindly provided by Karoly Fatyol. The *Rdr6* line was crossed with the 16C homozygous (hom) plants. The *rdr6*x16C plants were used to obtain the homozygous *rdr6*x16C lines. Homozygous *rdr6* verification was performed based on the reported flower phenotype [36] and Sanger sequencing of the RDR6 sequence. Homozygosity of the 16C-GFP allele was confirmed by the GFP-positive lines in the next generation. Although *rdr6* homozygous mutants were reported to be sterile, homozygous *rdr6*x16C plants gave rise to 1–2 fertile flowers per plant, which allowed us to amplify the homozygous *rdr6*x16C seeds.

### 2.7. Statistical Analysis

Statistical tests were performed by one-way ANOVA using JMP Software (version 16, SAS Institute, NC, USA). Unless otherwise stated, *p* < 0.05 was taken as the significance threshold.

### 2.8. Phylogenetic Tree

Nucleotide sequences of the Rep (or homologous) proteins of different geminiviruses were obtained from NCBI. MEGA (Molecular Evolutionary Genetics Analysis) was used for the phylogenetic tree, based on the neighbor-joining method.

## 3. Results

### 3.1. Phenotypic Analysis of Transient BCTIV Gene Expression and Silencing Inducers on N. benthamiana

Before we began investigating the VSR activity of BCTIV genes, via RT-PCR we confirmed that the binary vectors expressed the previously identified BCTIV genes (C1, C2, Rep, V1, V2, V3) upon agroinfiltration into 16C *N. benthamiana* leaves. (Appendix A) [1,37,38]. Notably, C1 and Rep expression also led to necrotic speckles starting from 5 dpi and the necrotic phenotype gradually got stronger in the following days as previously reported [35].

### 3.2. BCTIV-Rep and -V2 Suppressed GFP-FL Induced Local PTGS in 16C N. benthamiana

First, we aimed to investigate the role of BCTIV genes in the Sense-PTGS (S-PTGS) pathway. For this assay, full-length GFP (GFP-FL) was transiently expressed together with the BCTIV genes in stable GFP-expressing 16C *N. benthamiana* (Figure 1a). GFP-FL induced GFP expression under the control of the 35S promoter briefly before it acted as an S-PTGS inducer silencing itself and consequently silencing the stably expressed GFP at 5–6 dpi. Due to S-PTGS induction, GFP-FL agroinfiltrated leaf sectors turned red under UV light due to loss of GFP at 5–6 dpi. However, if a BCTIV gene functions as a viral suppressor of RNA silencing (VSR) the agroinfiltrated sector should remain green or become greener under UV light (Figure 1a). Evaluating the GFP expression level qualitatively under a UV lamp we first verified that TBSV-p19 had a strong VSR activity and GFP-FL control was acting as a silencing inducer, validating the experimental set-up (Figure 1b). We found that BCTIV-Rep and V2 but not C1, V1, and V3 had detectable VSR activity (Figure 1b, Appendix A). The quantification of the GFP signal using a fluorescent imager was performed by normalizing the non-vascular GFP intensity at the infiltrated zone to the non-vascular GFP expression outside of the infiltration zone in the same leaf. The normalized GFP intensity values showed that Rep and V2 had comparable levels of VSR activity, which were not as effective as TBSV-p19 (Figure 1c). The Western blot results also showed a higher accumulation of GFP protein after GFP-FL infiltration with V2 and Rep, when compared to the GFP-FL control (Appendix A). The intact GFP RNA level, which was monitored using a Northern blot assay with a GFP probe also showed a higher abundance of GFP transcripts in V2 and Rep-treated leaves, when compared to the GFP-FL control (Figure 1d). Similar to the GFP intensity measurements, intact GFP RNA levels in V2 and Rep were not as high as in the TBSV-p19 positive control (Figure 1d). We performed a quantitative RT-PCR using cDNA, synthesized by oligo-dT primers in order to compare the polyA-tagged intact GFP transcript levels and we found out that both V2 and Rep treatment yielded higher GFP-polyA transcript levels than the GFP-FL control and less than TBSV-p19 (Figure 1e). A small RNA Northern blot using GFP as a probe demonstrated a drop in the accumulation of siRNAs upon Rep and V2 treatment when compared to the GFL-FL control (Figure 1f).

### 3.3. Rep and V2 Did Not Interfere with the Spread of Silencing through Plasmodesmata

Local silencing induced by Sense-PTGS inducers like GFP-FL creates a visible red area encircling the infiltration zone, which shows stronger silencing than the infiltration zone (Figure 1a [39]). It was postulated that the silencing signal reaches this area via the short-range spreading of siRNAs through plasmodesmata [40,41,42]. We quantified the size of this silenced area by plotting the profile (using FIJI) along a line in the GFP channel. Although TBSV-p19 successfully blocked the formation of the silenced zone, neither Rep nor V2 hindered the transportation of the sRNAs to the neighboring cells compared to the GFP-FL control (Figure 1f–i). Also, the width of the silenced zone remained between 0.65–1.0 mm in Rep, V2, as well as the GFP-FL control treatments (Figure 1j–m).

### 3.4. Rep and V2 Suppressed Self PTGS of GFP-FL in Wild-Type N. benthamiana

The siRNAs produced after GFP-FL (S-PTGS inducer) infiltration could induce silencing by directing ARGONAUTE 1 (AGO1) to the transiently expressed GFP and to the stably expressed GFP RNA. Also, the quantification of the silencing was provided as a normalized value to the non-infiltrated areas. In order to measure the absolute GFP signal and monitor the self-silencing of the GFP-FL inductor, the GFP-FL and the BCTIV-Rep and -V2 were infiltrated into wild-type *N. benthamiana* leaves [34,43,44]. (Figure 2a). Both BCTIV-Rep and -V2 increased the GFP signal when compared to GFP-FL-only control. Similar to the infiltration experiments on 16C *N. benthamiana*, neither V2 nor Rep induced GFP expression nearly as high as TBSV-p19 (Figure 2b,c). Western blot, using GFP antibody, on the corresponding samples also showed that Rep and V2 led to higher GFP protein levels (Appendix A). The quantitative qRT-PCR using GFP-specific primers demonstrated an increase in the GFP amount upon Rep and V2 co-infiltrations when compared to the GFP-FL control (Figure 2d) but less than p19. A small RNA Northern blot using a probe against GFP showed a reduction in 21 nt- and 24 nt-sRNAs upon Rep, V2, and p19 treatment when compared to the GFP-FL control (Figure 2e).

### 3.5. Rep but Not V2 Suppressed Local Silencing Induced by a 5′GFP in 16C N. benthamiana

In order to monitor the effect of the S-PTGS inducer on the stable GFP expression, we used a 139 nt GFP fragment (5′GFP) mapping to the 5′ region of the GFP coding sequence, under the control of the 35S promoter (Figure 3a). Based on the GFP expression level under the UV light at 6 dpi, we found out that only the Rep protein could suppress the S-PTGS, while V2 did not show any sign of VSR activity (Figure 3b). For the exact quantification of the GFP levels after silencing induction and silencing suppression, the leaves were imaged in a high-resolution laser scanner (Figure 3b). The GFP intensity measurements in non-necrotic areas showed that the GFP expression was significantly (*p* < 0.05) higher in Rep-infiltrated 16C *N. benthamiana* leaves in comparison to 5′-GFP only or V2, 5′GFP co-infiltrated plants. (Figure 3c).

To avoid any artifacts due to necrotic areas and to reveal the mode of action of VSR activity of BCTIV-Rep, we investigated the GFP mRNA and small RNA populations upon co-infiltration in the leaf sections of 16C plants using Northern blot. Despite loading the same amount of material according to nanodrop measurements, the RNA quality of different co-infiltrations differed observably based on the rRNA bands on the agarose gel electrophoresis of the RNA samples. (Figure 3d). In particular, due to the necrotic areas caused by Rep, the RNA integrity was the lowest in the Rep + 5′-GFP co-infiltrated sample. Nevertheless, the GFP mRNA level in Rep + 5′-GFP was the highest when compared to V2, and the 5′GFP control but not as high as the untreated 16C sample (Figure 3d).

In addition, we performed an sRNA Northern blot on the same co-infiltrated samples. An end-labelled U6 snoRNA probe was used as a loading control, which did not reveal any major fluctuations in the sRNA concentration (Figure 3e). Samples from the untreated 16C did not show any sRNAs, while the 5′-GFP treated 16C samples had a clear accumulation of sRNAs, suggesting that the experimental setup was valid. The Rep + 5′-GFP sample showed a very faint sRNA band at the level of 21–24 nucleotides (nt), suggesting that Rep interfered with the silencing pathway, before the production of sRNAs by DCLs (Figure 3e). V2 co-infiltration with 5′-GFP led to a more limited decrease than Rep in the sRNA production in concordance with the previous literature [34].

### 3.6. PTGS Induced by a Hairpin Construct Targeting GFP Was Partly Repressed by Rep but Not by V2

Since Rep was capable of decreasing the sRNA production, necessary for S-PTGS, we addressed whether Rep could repress PTGS induced by hairpin RNA (GFP-IR) expressed under the control of the 35S promoter. Similar to the S-PTGS infiltrations using GFP-FL and 5′GFP, BCTIV-Rep and -V2 were co-infiltrated with GFP-IR in 16C plants (Figure 4a). GFP-IR is a very strong inducer of local silencing and at 6 dpi, BCTIV-V2 co-infiltrated leaves did not show loss of the silencing phenotype (Figure 4b). Nevertheless, Rep co-infiltrated with GFP-IR had a slightly higher level of GFP than GFP-IR only (Figure 4c), suggesting a partial suppression of GFP-IR-induced PTGS. Therefore, we investigated the molecular fingerprints of Rep and V2 on PTGS by checking GFP mRNA and sRNA levels using Northern blot on 16C *N. benthamiana*. Due to fewer necrotic areas in the Rep + GFP-IR sample, the RNA quality was comparable to the other treatments and controls. In addition, the abundance of GFP mRNA in 16C and the low level of GFP in the GFP-IR treatment validated the experimental setup. None of the treatments, including Rep, could fully block the decrease in the GFP mRNA level upon GFP-IR induction (Figure 4d).

Next, we performed a sRNA Northern blot on GFP-IR-induced PTGS samples. The U6 snoRNA loading control shows comparable amounts of total sRNAs among different treatments and controls (Figure 4e). In addition, GFP-IR only showed very strong sRNA accumulation, whereas no sRNAs were visible in the 16C control. Interestingly, we observed that the sRNA amount mapping to GFP was very low in the Rep + GFP-IR sample and slightly reduced in the V2 + GFP-IR sample at 6 dpi. (Figure 4e). It is noteworthy that the most abundant sRNAs upon GFP-IR induction were 24 nt siRNAs but for GFP-5′ and GFP-FL, the 21 nt siRNAs were more abundant than the 24 nt siRNAs (Figure 1f, Figure 2e and Figure 3e).

In the same line with the S-PTGS results, our data showed that the GFP-IR + GFP-FL co-infiltration created a silenced zone of 0.7–1.0 mm around the infiltration zone, which roughly corresponds to 15–20 cells (Appendix A). When GFP-FL and GFP-IR were co-infiltrated with V2, or Rep on 16C *N. benthamiana*, the local silencing phenotype was partially repressed at 5 dpi (Appendix A), yet did not block the formation of the silent red sector around the infiltration area (Appendix A).

### 3.7. Rep but Not V2 Suppressed Systemic Silencing

To find out whether the decrease in the sRNA amount upon expression of Rep and V2 correlated with systemic silencing in the 16C plants, the GFP-IR treated plants were observed and recorded regularly under UV light until 14 dpi (Figure 4f,g). Systemic silencing emerged at 10 dpi in plants infiltrated with GFP-IR only and upon V2 + GFP-IR co-infiltration the systemic silencing phenotype spread in the newly emerging leaves at 14 dpi, without any exception (Figure 4f). None of the plants co-infiltrated with Rep + GFP-IR (Figure 4g) showed systemic silencing, although the contribution of the necrotic effect of Rep to the absence of systemic silencing remains elusive.

### 3.8. Multiple Rep Domains Were Indispensable for VSR Activity

Rep protein has a large overlapping N terminus shared with the C1 protein. Splicing produces a frameshift at the C terminus of the C1, and the 14 nt between C1 and C2 is integrated into Rep. C2 constitutes the C terminus of Rep. C1 did not show VSR activity when it was co-expressed with GFP-FL on 16C leaves (Figure 5a,b, Appendix A). We investigated whether C2 could account for the VSR activity when it was co-expressed with GFP-FL in WT and 16C (Figure 5a,b). Based on the GFP intensity, C2 displayed a partial VSR activity both in the WT and 16C *N. benthamiana*, whereas C1 had no detectable VSR activity. Domain prediction of BCTIV-C1 by InterPro revealed the *Geminivirus* Rep catalytic domain, which contains four divalent metal cation binding residues. BCTIV-C2 did not show any predefined domains or structures. The frameshifted C1 and C1-C2 intergenic region in Rep created a homologous domain to the P-loop containing nucleoside triphosphate hydrolase (IPR027417), which was reported to be critical in viral DNA replication in *Tomato yellow leaf curl virus*. We divided each of C1 and C2 into three arbitrary sections and created twelve different truncated Rep proteins (Appendix A). All the Rep truncations were expressed together with either 5′-GFP (Appendix A) or GFP-FL (Appendix A) in 16C *N. benthamiana* plants. None of the Rep truncations recapitulated the Rep-VSR activity at 6 dpi which correlated with GFP quantification in the infiltrated leaf patches (Appendix A). It was noteworthy that the truncation of Rep with a 30 amino acid-long deletion at the C-terminus partially kept the VSR activity, suggesting that the end of the C terminus is dispensable for S-PTGS suppression (Rep4, Appendix A).

### 3.9. VSR Activity of Rep but Not of V2 Is RDR6 Dependent

RDR6 is a major regulator of S-PTGS. We addressed whether RDR is targeted by BCTIV-Rep or -V2 proteins using the *rdr6* mutant. We first tested whether the *rdr6*x16C still had leftover RNA-dependent-RNA polymerase activity by infiltrating S-PTGS inducers (GFP-FL, 5′GFP) and GFP-IR constructs. As expected, the 5′-GFP S-PTGS inducer, which requires RDR6 activity for silencing, did not lead to local silencing. We also observed that GFP-FL did not induce local silencing in *rdr6*x16C but the GFP induction remained mild. Of note, we anticipated a higher GFP expression in the *rdr6*x16C background comparable to the VSR positive control experiments in Figure 1b. On the other hand, GFP-IR, which does not require RDR6 activity for silencing, induced local silencing (Figure 6a) in *rdr6*x16C, despite the lack of a clear red area encircling the infiltration area.

Next, we co-infiltrated GFP-FL with V2, Rep, and TBSV-p19 to investigate how the VSR activity of these proteins relies on RDR6. Both p19 and BCTIV-V2 induced GFP expression in the *rdr6*x16C background (Figure 6b,c) to a level comparable to the increase in the 16C background, which contained functional RDR6 (Figure 1c). Therefore, BCTIV-V2 VSR activity was independent of RDR6. In contrast, the GFP signal intensity had a slight increase upon co-infiltration with Rep, which suggested that Rep VSR only partially was based on RDR6.

### 3.10. Rep and V2 Lead to Changes in the Expression of RNAi Genes

In order to investigate additional factors, which play a role in the VSR activity of BCTIV-Rep and V2, we conducted a quantitative qRT-PCR assay, monitoring the expression level of RNAi genes upon Rep and V2 infiltrations. The expression level of RNAi pathway genes upon Rep and V2 co-infiltration with GFP-FL was normalized to only GFP-FL control infiltration. For Rep co-infiltration, two time points, 4 dpi and 6 dpi, were selected to avoid the effect of necrosis. For V2 co-infiltrations, the leaves were harvested at 6 dpi. qRT-PCR results pointed out that the most drastic change upon Rep and V2 co-infiltrations was the downregulation of the AGO1 gene, which is indispensable for silencing (Figure 6d). DICER-LIKE 2 (DCL2) increased more than two-fold in Rep treatment at 4 dpi but remained unchanged at 6 dpi. The gene expression changes in other components of RNAi remained mild and statistically not significant.

## 4. Discussion

RNA silencing is a cellular defense mechanism conserved across kingdoms, which is triggered by dsRNA in plants. Despite being ssDNA viruses with no dsRNA phase in their replication cycle, geminiviruses are targeted by plants via post-transcriptional gene silencing by producing bi-directional transcription from opposite polarity or the strong fold-back structure of transcripts [13]. In addition, viral gene expression can lead to aberrant transcripts, which are further processed by RDRs to form dsRNAs. Therefore, geminiviruses must confront posttranscriptional gene silencing (PTGS) to achieve successful infections [45]. Several open reading frames were identified as PTGS suppressors in geminiviruses. However, the silencing suppressor activity of BCTIV-Rep was not extensively studied [46]. In this study, we used distinct approaches to dissect the role of BCTIV genes in the suppression of PTGS.

When the GFP-FL was agroinfiltrated on the WT plants, it induced GFP expression, but it also silenced itself due to the production of aberrant RNAs, triggering S-PTGS [47]. When GFP-FL was transiently expressed, BCTIV-V2 and -Rep worked as silencing suppressors (Figure 1 and Figure 2). However, when the GFP was solely expressed from the genome of 16C *N. benthamiana* (without transient GFP-FL expression) and the S-PTGS was induced by transient expression of 5′-GFP, BCTIV-V2 had no repressive effect on the silencing, whereas BCTIV-Rep substantially suppressed S-PTGS (Figure 3). The Northern blots on sRNA indicated that both V2 and Rep interfered with the sRNA production substantially upon S-PTGS induction by GFP-FL (Figure 1f and Figure 2e). However, Rep reduced the siRNA level more than V2 when S-PTGS was induced by GFP-5′ on the 16C background (Figure 3e).

GFP-IR is a very strong silencing inducer, which can lead to systemic silencing as early as 10 dpi. Upon transient expression of GFP-IR, due to the hairpin structure, dsRNA immediately forms and is subsequently processed by the DCL proteins. In this case, V2 could not repress the local silencing induced by GFP-IR, whereas Rep could partly reduce the silencing phenotype (Figure 4c). Nevertheless, the sRNA Northern blot showed that Rep could decrease the amount of sRNA produced upon GFP-IR expression. Although the decrease did not lead to suppression of local silencing, the systemic silencing phenotype was consequently affected. Contrary to Rep, V2 did not have any effect on systemic silencing. It has recently been proposed that the AGO proteins present at the locus of silencing induction sequester the sRNAs and mobile sRNAs can induce systemic silencing only when their concentration is above a certain threshold, determined by the concentration of AGO proteins [48]. Therefore, the reduction in sRNA level by Rep, which did not affect local silencing, could be low enough to avoid induction of systemic silencing. Additionally, it is worth mentioning that the necrotic phenotype after Rep infiltration could further decrease the production of sRNAs after 6 dpi, which contributed at least partly to the lack of the systemic silencing phenotype in these co-infiltrations.

Although this is the first report of the Rep protein of BCTIV as a silencing suppressor, Rep was previously identified as a PTGS suppressor in different geminiviruses like *Wheat dwarf virus* (WDV), a *Mastrevirus* [31,49]. However, the BCTIV-Rep protein is only very distantly related to the WDV-Rep proteins based on nucleic acid sequence (Appendix A). Therefore, the precise identification of the RNAi pathways that Rep proteins interfere with may shed light on uncharted functions of Rep proteins in geminiviruses. Additionally, Rep proteins from TYLCV, TGMV, and ACMV were shown to interfere with the plant DNA methylation machinery and suppress transcriptional gene silencing [50]. However, the role of the BCTIV-Rep protein on TGS remains unexplored. We infiltrated BCTIV proteins, including Rep into 16C plants transcriptionally silenced by VIGS (16C-TGS) from David Baulcombe’s lab [51] but none of the leaves regained detectable GFP expression. However, such an experiment only addresses the effect of BCTIV in the reversal of the DNA methylation not in the initiation or maintenance of DNA methylation. In addition to Rep, a recent publication also reported the local VSR activity of BCTIV-V2 [52], which overlapped with the results presented in this manuscript. Although no sequence homology has been established between *Becutrovirus*- V2 and *Begomovirus*-AV2, it is worth mentioning that -AV2 was identified as a suppressor with a weaker silencing outcome than P19 [53],

To highlight the mode of action of the PTGS suppressor activity of BCTIV, we hypothesized that the BCTIV-Rep protein interfered with RDR6 (Appendix A). In S-PTGS, where the WT and 16C *N. benthamiana* were inoculated with FL-GFP and 5′GFP, respectively, RDR6 was recruited to produce dsRNA from the aberrant GFP transcripts (Appendix A). The Rep protein suppressed the local silencing derived from RDR6-mediated dsRNA production. Therefore, 5′-GFP infiltrations did not lead to sRNA production when co-infiltrated with Rep. Unlike S-PTGS, in GFP-IR-induced PTGS, the hairpin transcript was already dsRNA, therefore RDR6 was not required to produce primary siRNAs, which lead to local silencing (Appendix A). Hence, BCTIV-Rep only partly suppressed IR-induced PTGS. It is noteworthy, however, that RDR6 was also indispensable for systemic silencing [54,55,56] and Rep but not V2 could block systemic silencing (Appendix A). Therefore, we investigated whether Rep interferes with RDR6 activity using *N. benthamiana* with a *rdr6* null mutation [36] and the results showed VSR activity of Rep was partly dependent on RDR6, whereas V2 VSR activity was independent of RDR6 (Figure 6b,c).

As evoked in the introduction, a key feature of S-PTGS is its strong dependency upon RDR6 activity, where the dsRNA products are mainly processed by DCL4 into 21nt siRNAs [57,58]. However, DCL4 activity was not regulated by BCTIV genes as shown by our results. Additionally, distinct RNAi gene regulation between the V2 and Rep infiltrations implied that Rep and V2 VSR activity may have multiple targets on RNAi, which can be at least partly distinct from each other [59,60,61] Moreover, the kinetics of initiation of VSR activity of Rep and V2 may also change due to differences in the timing of the deregulation of target genes and proteins.

As previously indicated, the Rep protein of WDV was functionally characterized. Truncations of WDV-Rep at the N terminus (9 amino acids) and the C-terminus (30 amino acids) did not affect the silencing suppression function and the N-terminus was critical for the necrotic phenotype [32]. We have further characterized BCTIV-Rep with truncating domains on both the N and the C terminus. We found that none of the truncated Rep proteins fully maintained the suppression of the silencing phenotype of BCTIV-Rep. Although C2 alone could partially lead to VSR activity, N terminal additions to the C2 protein yielded no VSR activity. One possible explanation for this is that there can be a functional domain also on the N terminus domain of C2. Additionally, C-terminus deletions but not N-terminus deletions of Rep truncations showed necrotic phenotypes comparable to the full-length Rep protein. These results suggest that multiple domains of BCTIV-Rep except the very end of the C-terminus, are indispensable for the silencing function, and the N-terminus of the Rep protein is essential for necrosis, which is similar to the wheat dwarf virus (WDV) Rep protein [31].

## 5. Conclusions

In this study, we determined the BCTIV-Rep and -V2 as viral suppressors of PTGS. We showed that Rep inhibits local and systemic silencing partly in an RDR6-dependent manner. BCTIV-V2 however, has S-PTGS suppression activity independent of RDR6.

## Figures and Tables

**Figure 1 viruses-15-01996-f001:**
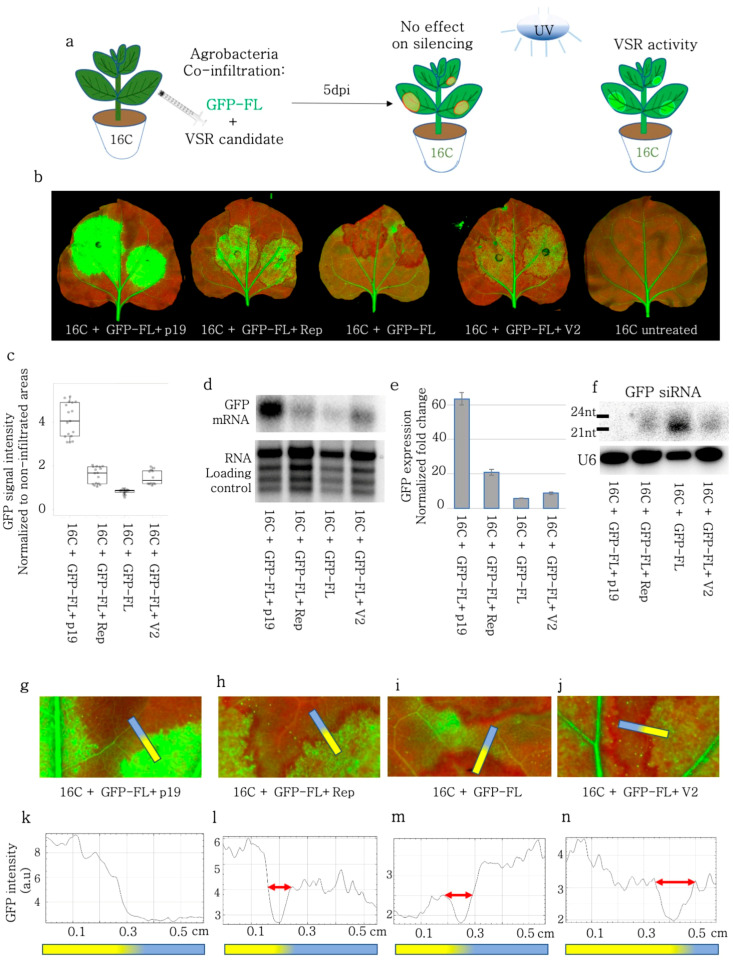
VSR activity of BCTIV-Rep and -V2 genes on GFP-FL induced S-PTGS on 16C *N. benthamiana*. (**a**) GFP expressing 16C *N. benthamiana* leaves were co-infiltrated with S-PTGS inducer GFP-FL together with each of BCTIV-Rep, BCTIV-V2 and TBSV-p19, as positive control and GFP-FL alone as the negative control. At 6 dpi, if GFP-FL leads to red sectors under the UV lamp, this suggests that VSR activity of the co-infiltrated gene is negligible. However, if the infiltrated sector remains green or becomes greener under UV lamp, it shows VSR activity for the gene of interest. (**b**) High-resolution images of the leaves at 6 dpi were obtained at 50 µm/pixel using green and far-red channels of PharosFX™ imager. Green (GFP) and far-red channels (chlorophyll) were merged (**c**) Non-vasculature and non-necrotic sectors of the high-resolution leaf images were quantified based on GFP via dividing the GFP intensity in the infiltrated area by the average GFP intensity of the non-infiltrated area. At least three leaves and multiple areas per leaf were quantified. One-way ANOVA statistical test showed significance at *p* < 0.05. (**d**) Top panel shows Northern blot results using a radioactively labelled GFP probe. GFL-FL induced GFP silencing and lowered GFP transcript levels and p19 was used as a positive control, showing high VSR activity. rRNA served as loading control, for agarose gel electrophoresis and ethidium bromide staining. (**e**) The qRT-PCR using GFP and PP2A genes was performed, using RNA material from the infiltrated sections on four replicates. GFP expression was normalized to the PP2A expression to calculate the normalized fold change. (**f**) Top panel shows a small RNA Northern blot using a radioactively labelled GFP probe. The bands correspond to 21 and 24 nt long siRNAs. GFP-FL produced a high concentration of siRNA. The lower panel is the small RNA Northern blot, obtained by a radioactively labelled U6 RNA probe as the control for RNA loading. U6 RNA labelling took place on the same membrane shown in the panel above, after stripping. (**g**–**j**) Close up images of the borders of infiltrated areas. Yellow-blue gradient bars indicate infiltrated areas (yellow) and non-infiltrated areas (blue). (**k**–**n**) The GFP channel intensity plot across the border of the infiltration area. *Y*-axis indicates GFP intensity in arbitrary unit (a.u) and *x*-axis shows the distance of a given spot from the end of yellow infiltrated area. The red arrows were drawn to highlight the length of the red-silenced sectors, which encircled the infiltrated area.

**Figure 2 viruses-15-01996-f002:**
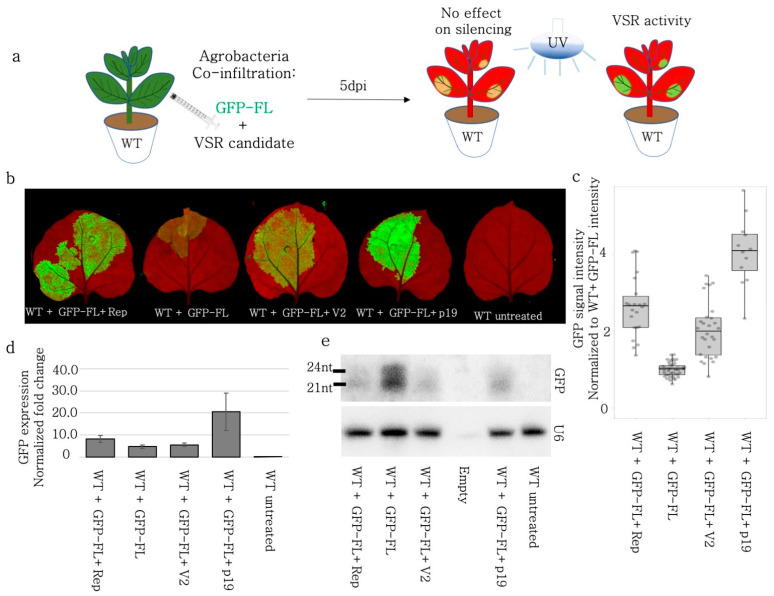
BCTIV-Rep and -V2 genes repressed S-PTGS upon transient expression of GFP-FL on wildtype *N. benthamiana*. (**a**) S-PTGS-inducing GFP-FL was co-infiltrated with BCTIV-Rep, BCTIV-V2, TBSV-p19 (as a positive control), and alone as a negative control on WT *N. benthamiana*. Upon VSR activity, GFP-FL agroinfiltration was expected to show higher GFP fluorescence than GFP-FL alone under UV light. (**b**) High-resolution images of the leaves infiltrated by GFP-FL and virus genes were obtained at 50 µm/pixel using green and far-red channels of PharosFX™ imager. Green (GFP) and far-red channels (chlorophyll) were merged using FIJI. (**c**) Infiltrated areas in the leaves were quantified based on GFP intensity. At least three leaves and multiple areas per leaf were quantified. Absolute GFP intensity values were divided by average GFP intensity of the GFP-FL alone treatment. One-way ANOVA statistical test shows significance at *p* < 0.05. (**d**) The qRT-PCR for GFP and PP2A genes were performed on four replicates, using RNA material from the infiltrated sections. GFP expression was normalized to the PP2A expression to calculate “normalized fold change”. All three conditions showed higher GFP expression when compared to GFP-FL only (*p* < 0.05). (**e**) Top panel shows a small RNA Northern blot using a radioactively labelled GFP probe. The bands correspond to 21 and 24 nt long siRNAs. GFP-FL serves as a highly abundant siRNA control. The lower panel is the small RNA Northern blot by using U6 RNA labelling as a control for RNA loading. U6 RNA labelling took place on the same membrane shown in the panel above, after stripping.

**Figure 3 viruses-15-01996-f003:**
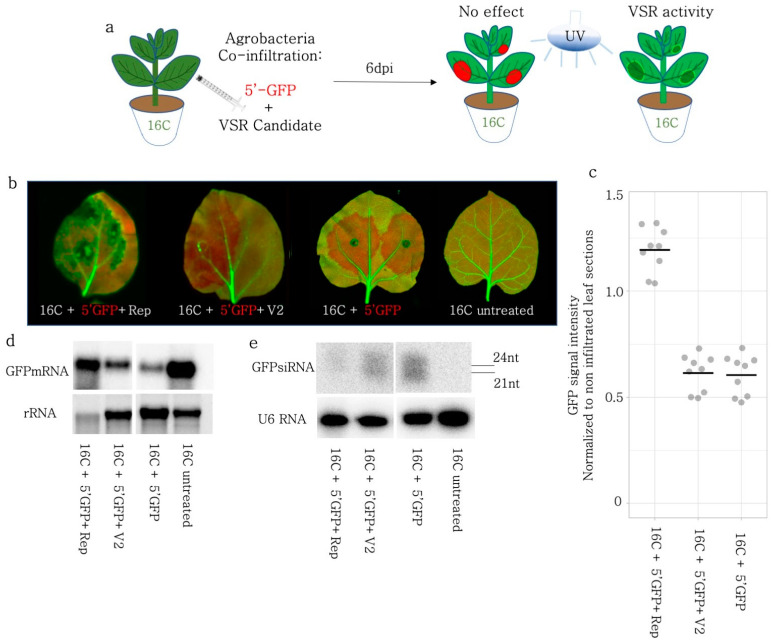
VSR activity of BCTIV-Rep and -V2 genes on 5′-GFP induced S-PTGS on 16C *N. benthamiana*. (**a**) GFP-expressing 16C *N. benthamiana* leaves were co-infiltrated with S-PTGS inducer 5′-GFP and BCTIV-Rep and BCTIV-V2. At 6 dpi, if 5′-GFP leads to red sectors under the UV lamp, it suggests that the VSR activity of the co-infiltrated gene is negligible. However, if the infiltrated sector remains green under a UV lamp, it shows VSR activity for the gene of interest. (**b**) High-resolution images of the leaves infiltrated by 5′-GFP together with Rep, V2, and 5′ GFP alone were obtained at 50 µm/pixel using green and far-red channels of PharosFX™ imager. Green (GFP) and far-red channels (chlorophyll) were merged. (**c**) Non-vasculature sectors of the high-resolution leaf images were quantified based on GFP and normalized to the non-infiltrated parts. At least three leaves and multiple areas per leaf were quantified. One-way ANOVA statistical test showed significance at *p* < 0.05. (**d**) Top panel shows Northern blot results using a radioactively labelled GFP probe. 5′-GFP serves as a low GFP expression control and a 16C-untreated sample shows high-abundance GFP mRNA control. rRNA is a loading control for the upper panel, after agarose gel electrophoresis and ethidium bromide staining. (**e**) Top panel shows small RNA Northern blot using a radioactively labelled GFP probe. The bands correspond to 21 and 24 nt long siRNAs. 5′-GFP serves as a highly abundant siRNA control and 16C-untreated serves as a clean negative control. The lower panel shows the Northern blot with U6 RNA probe as a control for RNA loading. U6 RNA labelling took place on the same membrane given in the panel above, after stripping.

**Figure 4 viruses-15-01996-f004:**
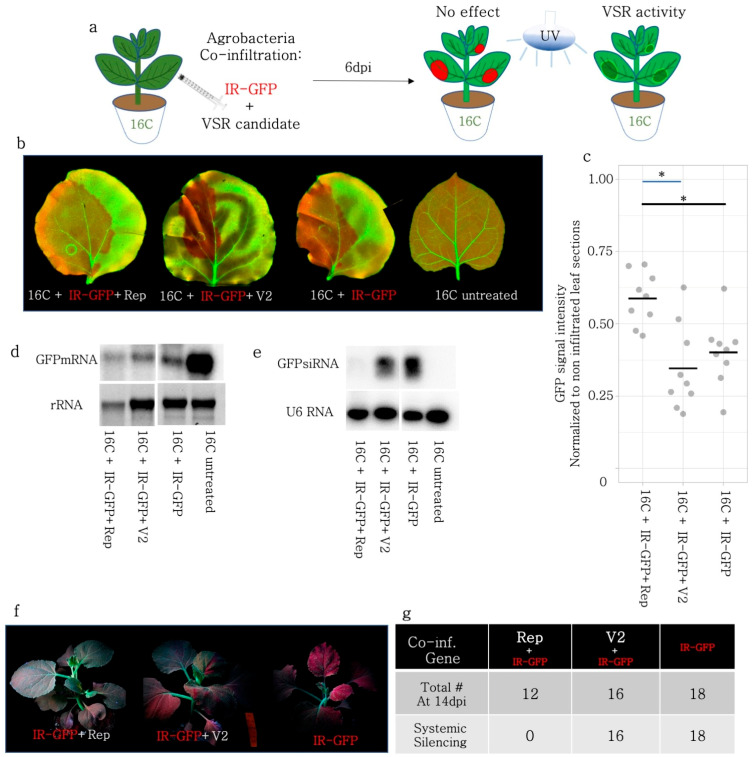
PTGS induced by GFP-IR was partially blocked by Rep and could not be suppressed by V2. (**a**) PTGS-inducing GFP-IR was co-infiltrated with Rep, and V2 of BCTIV on 16C *N. benthamiana*. Upon VSR activity, GFP-IR agroinfiltrated sectors are expected to retain high GFP fluorescence under UV light. (**b**) High-resolution images of the leaves infiltrated with GFP-IR together with Rep, V2, and GFP-IR only were obtained at 50 µm/pixel using green and far-red channels. (**c**) Non-vasculature sectors of the high-resolution leaf images were quantified based on GFP intensity. GFP intensity of the infiltrated area was normalized to the average GFP intensity of the non-infiltrated area. At least three leaves and multiple areas per leaf were quantified. The normalized GFP intensity of the V2 co-infiltration was not significantly different from the GFP-IR control but Rep treatment yielded higher GFP intensity when compared to GFP-IR only or V2 (* *p* < 0.05). (**d**) Top panel shows Northern blot results using a radioactively labelled GFP probe. GFP-IR serves as a low GFP expression control and a 16C-untreated sample as a high-abundance GFP mRNA control. rRNA was used as a loading control for the upper panel, after agarose gel electrophoresis and ethidium bromide staining. (**e**) Top panel shows a small-RNA-Northern blot using a radioactively labelled GFP probe. The bands correspond to 21 and 24 nt long siRNAs. GFP-IR serves as a highly abundant siRNA control and 16C-untreated as a negative control. The lower panel shows the small RNA Northern blot by U6 RNA probe as a control for RNA loading. U6 RNA labelling took place on the same membrane given in the panel above, after stripping. (**f**) A minimum of twelve 16C *N. benthamiana* used for GFP-IR co-infiltration with VSR candidates are kept until 14 dpi. The images of the whole plant are taken using a DSLR Camera under UV light. (**g**) The total number of co-infiltrated plants, remaining healthy at 14 dpi is provided on the top row, and the number of plants showing systemic silencing in at least two leaves is given the bottom row.

**Figure 5 viruses-15-01996-f005:**
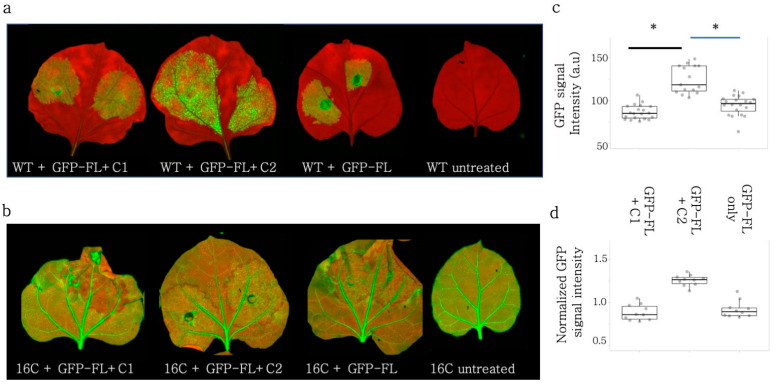
C2 subunit of Rep can partially account for the VSR activity of Rep but C1 subunit has no VSR activity. GFP-FL is co-infiltrated with BCTIV-C1, BCTIV-C2, and alone as the negative control on (**a**) Wt *N. benthamiana*. (**b**) on 16C *N. benthamiana*. High-resolution images of the leaves were obtained at 50 µm/pixel using green and far-red channels of PharosFX™ imager. Green (GFP) and far-red channels (chlorophyll) were merged using FIJI. (**c**) Non-vasculature sectors of the high-resolution leaf images were quantified based on GFP intensity (arbitrary unit). At least three leaves and multiple areas per leaf were quantified. GFP increase in Rep was significantly increased (*p* < 0.05) when compared to C1 or GFP-FL only control. (**d**) Non-vasculature sectors of the high-resolution leaf images were quantified based on GFP and normalized to the non-infiltrated parts. At least three leaves and multiple areas per leaf were quantified. GFP decrease in Rep was significantly less (*, *p* < 0.05) when compared to C1 or the GFP-FL control.

**Figure 6 viruses-15-01996-f006:**
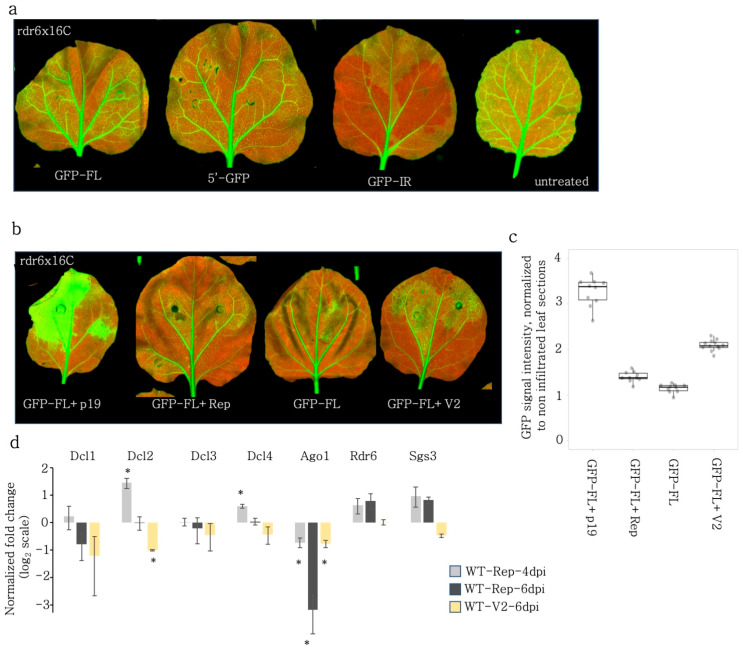
Distinct mechanisms regulate VSR activity of Rep and V2. (**a**) GFP-FL, 5′GFP, and GFP-IR were infiltrated to assess the PTGS activity in rdr6x16C line. The S-PTGS inducers GFP-FL and 5′GFP failed to induce silencing on rdr6 mutant background. The PTGS inducer GFP-IR induced silencing in the absence of functional RDR6. (**b**) GFP-FL was co-infiltrated with TBSV-p19, BCTIV-Rep, -V2, and alone into rdr6x16C leaves. High-resolution images of the leaves 5 dpi were obtained at 50 µm/pixel using green and far-red channels of PharosFX™ imager. Green (GFP) and far-red channels (chlorophyll) were merged using FIJI. (**c**) Non-vasculature sectors of the high-resolution leaf images were quantified based on GFP intensity normalized to the non-infiltrated sectors. At least three leaves and multiple areas per leaf were quantified. One-way ANOVA statistical test showed significance at *p* < 0.05. (**d**) qRT-PCR was performed on wt *N. benthamiana* co-infiltrated with silencing inducer and VSR genes at the given time points for the RNAi genes given in the *x*-axis. The Cq values of the given genes were normalized to the PP2A expression to find relative gene expression. Then for each Rep and V2 treatment, the relative gene expression was normalized to the relative gene expression values in GFP-FL infiltrated wt *N. benthamiana* to find the enrichment values, which were plotted in log2 scale. Value 0 baseline indicates no change in RNAi gene expression. Statistical significance is calculated with Student’s *t*-test between the given sample and GFP-FL control (*, *p* < 0.05).

## Data Availability

The coding sequences of the BCTIV genes are provided in the text. In addition, plasmids used in this study are available upon request.

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
