# Peer review of "Beet Curly Top Iran Virus Rep and V2 Suppress Post-Transcriptional Gene Silencing via Distinct Modes of Action"

_viruses, 2023, doi:10.3390/v15101996_

Round 1
Reviewer 1 Report
Interesting work worthy of publication. Presentation of results was confusing.
Line 115 - Capitalize Agrobacterium
Line 119 - fix spelling of Agrobacterium
Line 125 - Capitalize and italicize Agrobacterium; change 'has been' to 'was'
Line 174 - Capitalize 'Sanger'
Line 195 - add (data not shown)
Supplement Figure 1 - fix spelling of Becurtovirus
Supplement Figure 1 legend - change to '..similarity of the C-strand of becurtovirus..'
Figure 6 - This must be cited and discussed in the Results section. Currently this is only found in the discussion. It must first be included in the Results.
Figure 6 legend - change all verbs to past tense
Supplemental Figures - change spelling from Agrobacteria to Agrobacterium
Make sure verbs are used in past tense, particularly in the Figure 6 legend.
Author Response
We would like to thank the reviewer #1 for pointing out the language mistakes. We corrected all the points, given by reviewer #1.
We also agree with the reviewer that the presentation of the results had room for improvement. For this purpose, we modified the Figure legends, Supplementary Figures 1, 5, 6, 9, and 10.
Upon the reviewer's suggestion, we moved Figure 6 and the details of the experiments to the end of the results.
We hope the new version of the manuscript and the figures will fully address the reviewer's concerns.
Reviewer 2 Report
In this manuscript the authors investigated the ability of Rep and V2 proteins of BCTIV to suppress the PTGS machinery via distinct modes of action. Overall the MS is well written, the results are adequately presented and are supported by the data presented here. My main comment is that there is no description of the results obtained during the experiments with the rdr6x16c plants (figure 6). However, there is an extended description of these results in the "Discussion" section. The authors should add a paragraph in the "Results" section and limit their extent in "Discussion". Some minor comments can be found in the attached file.

Overall, minor editing is needed. My only comment is that quite frequently the authors write on both past and present tense in one sentence. They should decide (preferably past tense) and correct accordingly throughout the manuscript.
Author Response
We would like to thank the reviewer for providing a PDF file highlighting the language and semantic mistakes. We corrected all of them. Especially we paid special attention to the use of present/past tense.
In addition, we moved Figure 6 and the details of the experiments to the end of the results. We still kept a small discussion highlighting the reasoning behind the experiment and the potential implications.
We hope the new version of the text addresses all the concerns of the reviewer and it is more understandable and clear than the previous version.
Round 2
Reviewer 1 Report
All of the suggestions were accepted and changes made. Minor spelling and editing are needed as marked below.
Line 113 - fix spelling of fragments
Line 147 - repeated word - and and
Lines 204,271,305,418,495,498,566 - insert space after N. - N. benthamiana
Line 368 - fix spelling of labelling
Minor editing needed.
Author Response
All the corrections suggested by the reviewer have been done. We thank the reviewer for carefully reading the manuscript.